# The TOX2 Gene Is Responsible for Conidiation and Full Virulence in *Fusarium pseudograminearum*

**DOI:** 10.3390/cimb47090714

**Published:** 2025-09-02

**Authors:** Sen Han, Shaobo Zhao, Yajiao Wang, Qiusheng Li, Mengwei Sun, Lingxiao Kong, Xianghong Chen, Jianhai Gao, Yuxing Wu

**Affiliations:** 1Plant Protection Institute, Hebei Academy of Agriculture and Forestry Sciences, Baoding 071000, China; hansen19920603@163.com (S.H.); yajiaowang515@163.com (Y.W.); alidd@163.com (Q.L.); 17330283350@163.com (M.S.); konglingxiao163@163.com (L.K.); 13931226235@163.com (X.C.); 2Key Laboratory of Integrated Pest Management on Crops in Northern Region of North China, Ministry of Agriculture and Rural Affairs, Baoding 071000, China; 3IPM Innovation Center of Hebei Province, Baoding 071000, China; 4International Science and Technology Joint Research Center on IPM of Hebei Province, Baoding 071000, China; 5Plant Protection and Quarantine General Station of Hebei Province China, Shijiazhuang 050000, China; zhaoshaobo@163.com; 6Bureau of Agriculture and Rural of Cangxian, Cangzhou 061000, China; cxnyjyqb@163.com

**Keywords:** *Triticum aestivum*, *TOX2*, conidiation, secondary metabolite, Fusarium crown rot

## Abstract

Fusarium crown rot, a widespread and destructive disease affecting cereal crops (particularly wheat and barley), is primarily caused by the soil-borne fungal pathogen *Fusarium pseudograminearum*. Secondary metabolites (SMs) play a crucial role in colonization and host tissue invasion by pathogenic fungi. In this study, we investigated the functional role of *FpTox2*, a secondary metabolite-related gene in *F. pseudograminearum*. An *FpTox2* deletion mutant exhibited significantly reduced radial growth compared to wild-type *F. pseudograminearum*. Notably, the mutant strain completely lost conidiation capacity under induced conditions. Furthermore, although it showed decreased sensitivity to the cell membrane inhibitor sodium dodecyl sulfate (SDS), the mutant demonstrated enhanced susceptibility to NaCl, a metal ion stressor. Most importantly, the pathogen’s virulence was markedly attenuated in wheat stem base infections following *FpTox2* deletion, and we demonstrated that *FpTox2* regulates pathogen virulence by influencing deoxynivalenol production. In conclusion, *FpTox2* is crucial for vegetative growth, asexual development, abiotic stress responses, and full virulence in *F. pseudograminearum*.

## 1. Introduction

Wheat, one of the world’s most important staple crops, faces numerous disease threats during its growth cycle [1]. Fusarium crown rot (FCR) is a particularly significant disease that mostly affects the stem bases of wheat plants. This soil-borne disease, caused primarily by *Fusarium pseudograminearum*, among other pathogens [2,3], leads to stem tissue decay that disrupts normal plant development [2,4,5]. The disease initially manifests as water-soaked lesions with discoloration, which spread progressively throughout the stem and cause tissue softening and rot [6]. Severe infections can result in complete plant death, significantly compromising wheat yield and quality. *F. pseudograminearum*, a highly pathogenic fungus prevalent in agricultural soils, causes various crop diseases with substantial economic impacts [3,7]. Research data indicate that FCR, caused by *F. pseudograminearum*, is prevalent in the wheat-growing regions of Australia and the Pacific Northwest of the United States [7,8]. The disease typically results in a 10–35% reduction in wheat grain yield [7]. It is estimated that the Australian wheat industry incurs annual losses of AUD 88 million due to FCR, with the potential losses soaring to 434 million under favorable disease conditions [7]. Notably, FCR has emerged as a major disease in China’s Huang-Huai wheat-growing region, with its severity showing a sustained upward trend in recent years [9,10]. In parallel, there has been growing global research interest in *F. pseudograminearum* in wheat-growing regions, establishing it as a key focus in agricultural biology [11]. This pathogen primarily targets crop roots and stems, disrupting plants’ cellular structures and causing diseases that reduce or completely destroy their yields. Its pathogenic mechanisms involve both direct invasion and toxin production. Through secretion of various secondary metabolites (SMs), *F. pseudograminearum* degrades plant cell walls and membranes, leading to cellular dysfunction or death [12].

Fungal secondary metabolites (SMs) provide fungi with a key evolutionary advantage [13]. These low-molecular-weight natural products, derived from primary metabolic precursors, are non-essential for growth like primary metabolites [13,14,15] yet play crucial roles in ecological competition, host infection, and fungi’s impacts on human, animal, and plant health [13,16]. Fungi have evolved diverse SM production capabilities in response to environmental pressures. These compounds serve multiple biological functions as virulence factors, chemical defense mechanisms, insect attractants, stress protectants, developmental regulators, and inter-organism communication signals [14]. Their synthesis is environmentally regulated [14], resulting in remarkable chemical and functional diversity. Fungal SMs include both beneficial compounds (e.g., penicillin, lovastatin) and harmful toxins (e.g., aflatoxin, fumonisin) [13,17]. Certain SMs, like pigments, polyols, and mycolins, enhance pathogenicity and stress tolerance against temperature, UV, and oxidative challenges [18]. Necrotrophic fungi, which have broader host ranges than biotrophs, employ cell wall-degrading enzymes and toxic SMs or peptides [14,19]. *Fusarium*, a hemibiotrophic fungus, transitions to necrotrophy by producing toxins and cellulolytic enzymes that kill host tissue for nutrient absorption, enabling its survival across growth stages and nutritional conditions. *F. graminearum*, a globally significant pathogen of cereals (wheat, maize, barley, oats), produces multiple mycotoxins, including deoxynivalenol (DON), zearalenone (ZEA), and fusarin C [20,21]. Notably, the production of DON derivatives correlates with the pathogenicity of *F. graminearum* [22,23] and crop yield/quality reduction. Nitrogen starvation specifically triggers the trichothecene pathway, inducing the biosynthesis of DON [24,25,26], a potent virulence factor causing substantial agricultural losses and human health risks [27].

Killer toxins represent a specialized class of secondary metabolites that disrupt cellular function by inhibiting voltage-gated calcium channels and interfering with calcium-dependent signaling pathways in target organisms [28,29,30,31]. Among these is the KP4 killer toxin, a virulence protein secreted by pathogenic microorganisms to suppress growth and induce cell death [28]. One of three virally encoded proteins (KP1, KP4, KP6) originally identified in corn smut strains, KP4 exhibits unique specificity against *Ustilaginales* with minimal sequence similarity to KP1/KP6 [32,33,34]. Fungal genomes contain over 200 KP4-like (KP4L) homologs, which are predominantly present in Ascomycetes (79), with fewer in *Basidiomycetes* (5) and *Zoopagomycota* (1). Ascomycete KP4L proteins primarily occur in *Sordariomycetes* and *Eurotiomycetes* [33]. A comparative analysis of 12 *Fusarium* isolates revealed KP4L/UmKP4 structures conserved between Ascomycetes and Basidiomycetes, suggesting functional conservation [28,32]. Notably, *Fusarium graminearum* possesses three KP4-encoding genes upregulated during wheat infection and scab development, implicating their pathogenic role [32]. KP4L proteins serve dual functions: as virulence factors during plant infection and as mediators of fungal competition by enhancing antifungal compound penetration [35,36,37,38,39]. This competitive mechanism resembles yeast killer toxin systems that dominate ecological niches [28,40]. Transcriptomic studies of *F. graminearum*–*Trichoderma* gamsii interactions demonstrate KP4L’s antagonistic function, with its expression levels inversely correlating with inter-fungal distance [28,38,39]. The *Fgkp4l* gene appears to be particularly crucial during close-range fungal antagonism [39]. Remarkably, heterologous KP4 expression confers plant resistance against smut fungi and select Ascomycetes (e.g., *Alternaria alternata*, *Phoma exigua* var. *exigua*) in maize, tobacco, and wheat [41,42,43,44].

A comprehensive understanding of the KP4 protein’s pathogenic mechanism in *Fusarium* species is crucial for elucidating its biological functions and establishing the theoretical foundations for developing effective disease control strategies. In this study, we characterized FpTox2, a previously unstudied KP4-like (KPL4) homologous protein in *F. pseudograminearum*. Our growth rate analysis demonstrated FpTox2’s essential role in *F. pseudograminearum* development. Further investigations revealed that FpTox2 positively regulates both conidiation and the expression of *TRI5*, a key gene involved in trichothecene biosynthesis. These findings indicate that FpTox2 contributes to pathogenicity by influencing deoxynivalenol (DON) production and conidial germination processes. Thus, the KP4 homolog FpTox2 modulates the virulence of *F. pseudograminearum* through dual regulatory mechanisms: controlling asexual reproduction and coordinating mycotoxin biosynthesis pathways.

## 2. Methods

### 2.1. Strains and Growth Conditions

The wild-type strain 2035, identified as *F. pseudograminearum* through an ITS primer analysis, was collected from the field and preserved at the Laboratory of Fungal Diseases, Institute of Plant Protection, Hebei Academy of Agriculture and Forestry Sciences. The wild-type, Δ*FpTox2*, and Δ*FpTox2*-C strains used in this study were cultured on potato dextrose agar (PDA, containing 20% potato extract, 2% glucose, and 1.5% agar) at 25 °C.

### 2.2. Genetic Methods

In-frame deletions in *F. pseudograminearum* were achieved through double-crossover homologous recombination following a previously described method [45]. Using genomic DNA from *F. pseudograminearum* strain 2035 as a template, we amplified the upstream and downstream sequences of *FpTox2* with the primer pairs *FpTox2*-1F/2R and *FpTox2*-3F/4R. The hygromycin phosphotransferase (HPH) selection marker was amplified from a pFL2 plasmid using HYD-F/R primers. These three fragments were subsequently fused using the double-joint PCR method. The resulting PCR product was transformed into *F. pseudograminearum* 2035 protoplasts following standard protocols. Transformants were initially selected on PDA plates supplemented with 100 μg/mL of hygromycin. The presence of putative transformants was then confirmed through PCR and Southern blot analyses to verify successful *FpTox2* gene replacement.

For complementation, we amplified the *FpTox2* coding sequence along with its promoter region from wild-type strain 2035 genomic DNA using *FpTox2*-9F/10R primers. The PCR product was co-transformed with an XhoI-digested pFL2 vector into the yeast strain XK1-25 for homologous recombination. The recombinant plasmid was subsequently isolated, transformed into Escherichia coli DH5α for amplification, and verified through sequencing. The confirmed plasmid was then introduced into the *FpTox2* deletion mutant via protoplast transformation. Transformants were selected on a TB3 medium (0.3% yeast extract, 0.3% casamino acids, 20% sucrose, and 1.5% agar) and validated via a PCR using *FpTox2*-5F/6R primers. The primers used in this assay are listed in Appendix A.

### 2.3. Analysis of Growth Rate, Conidiation, and Conidial Germination Rate

For the growth rate and conidiation assays, the tested strains were first activated on a PDA medium for 3 days. A 5 mm mycelial plug was then aseptically transferred from the edge of the colony to a fresh PDA medium. The colony diameters were measured after 3 and 5 days of incubation at 25 °C in complete darkness. To assess conidiation and conidial germination rates, 5 mm mycelial plugs were inoculated into 50 mL of a carboxymethyl cellulose (CMC) medium in triangular flasks and incubated at 25 °C with shaking at 175 rpm for 3 or 5 days. The conidial production was quantified using a hemocytometer. For germination rate determination, conidial suspensions were standardized to equal concentrations, and 20 μL aliquots were spread evenly on 0.2% glucose agar plates using sterile glass rods. After 6–8 h of incubation at 25 °C in darkness, a Leica Microsystem was used to observe the germination rate of the spores. When the spore germination rate of the wild-type strain exceeded 90%, the germination rates of the Δ*FpTox2* and Δ*FpTox2*-C strains were statistically analyzed. All experiments included three biological replicates and were repeated three times independently.

### 2.4. Analysis Utilization of Different Carbon Sources

To compare the carbon source utilization between the wild-type and mutant strains, mycelial plugs from each strain were inoculated onto a YSS medium supplemented with six different carbon sources (0.05 M of glucose, glycerol, sodium malate, sodium succinate, sodium acetate, or sodium citrate). The cultures were incubated at 25 °C for 3 days, after which colony diameters were measured and compared between strains.

### 2.5. Stress Assay

To analyze the role of FoTox2 in the response to abiotic stress, strains were cultured on PDA for 3 days before being transferred to a CM medium containing stress agents: H_2_O_2_ (oxidative stress), NaCl/CaCl_2_ (osmotic stress), SDS (membrane disruptant), and Congo Red (CR; cell wall inhibitor). For each strain/mutant, a 5 mm mycelial plug from the colony periphery (3-day-old culture) was inoculated onto a test plate. After 72 h of incubation at 25 °C in darkness, the colony diameters were measured along two perpendicular axes, with the radial growth calculated by subtracting the initial plug diameter (5 mm). Three biological replicates were performed.

### 2.6. Plant Infection Assay

The virulence of the wild-type and mutant strains against wheat coleoptiles was assessed following a previously established protocol [10]. Seeds of the susceptible wheat cultivar Shixin 828 were surface-sterilized with 3% sodium hypochlorite for 3 min, rinsed three times with sterile distilled water, and then germinated on water agar. Subsequently, 5 cm-long wheat seedlings were inoculated with mycelial plugs of both wild-type and mutant strains and incubated under controlled conditions of 25 °C, 60% humidity, and an adjustable light level. The disease severity was assessed 7 days post-inoculation using a 0–4 scale based on the proportion of coleoptiles infected by *F. pseudograminearum*, employing the following grading criteria: grade 0: no infection; grade 1: slight infection (0–25%); grade 2: moderate infection (25–50%); grade 3: severe infection (50–75%); and grade 4: extremely severe infection (75–100%).

### 2.7. Determination of DON Production

For the analysis of DON production, mycelial plugs from each tested strain were inoculated into 100 mL of a CMC medium in Erlenmeyer flasks and incubated with shaking (175 rpm) for 5 days. The resulting conidia were harvested, washed three times with sterile distilled water, and resuspended to a final concentration of 1 × 10^6^ conidia/mL. A 10 μL aliquot of each conidial suspension was then transferred to 30 mL of a trichothecene biosynthesis induction (TBI) medium in a fresh Erlenmeyer flask and cultured under shaking conditions (175 rpm) for 7 days. Following incubation, the mycelial biomass was removed through filtration, and the filtrate was diluted 50-fold. The DON concentrations were quantified using a commercial detection kit (230720-OZ, Wise Science and Technology Development Co., Ltd., Suzhou, China). This kit was based on the competitive ELISA principle, using microplates pre-coated with a deoxynivalenol (DON) antigen. After adding the samples/standards and antibodies, the DON in the samples competed with the antigen to bind to the antibodies. Unbound antibodies were removed through washing, followed by the addition of an enzyme-labeled secondary antibody and chromogenic substrate (which turned blue, then yellow after termination). The absorbance was measured at 450 nm, with a negative correlation between the absorbance and toxin concentration. Quantitative results were obtained by referencing the standard curve. The experiment included three biological replicates per treatment and was independently repeated three times [46].

### 2.8. RNA Extraction and qRT-PCR Assay

The mycelia of each strain/mutant were collected and washed after 12 h of incubation in flasks containing 100 mL of yeast extract peptone dextrose broth (YEPD) medium. RNA was extracted using an RNA simple total RNA kit (DP441, Tiangen Biochemical technology Co., Ltd., Beijing, China) according to the manufacturer’s protocols. A quantitative real-time reverse transcription PCR (qRT-PCR) was carried out using the *Fptef1* gene as an internal control, as described previously [47]. The primers used in this assay are listed in Appendix A.

## 3. Results

### 3.1. Identification of Tox2 Gene and Mutant Construction

The FpTox2 protein (listed in Appendix A), which contains 125 amino acids (aas), was identified as being 99.2% homologous with the *Fusarium graminearum* KP4 domain-containing protein (Figure 1A). To construct the *FpTox2* gene deletion mutant, we replaced the entire open-reading frame of the *FpTox2* gene with the hygromycin B gene via homologous recombination. The up- and downstream regions of the *FpTox2* gene were linked to the hygromycin B gene using a double-joint PCR approach. Then, the *FpTox2* deletion constructs were transferred into the protoplast of the *F. pseudograminearum* wild-type strain 2035, and the transformed strain was verified via PCR amplification. From the wild-type strain, a 313bp-long fragment could be obtained, but the same fragment could not be amplified in Δ*FpTox2* (Figure 1B). Then, we designed a probe, h, of the hygromycin B gene to verify whether these were the target bands in the wild-type strain 2035 and Δ*FpTox2* genomes, which were digested by the XbaI enzyme using Southern blot tests. The Southern blot results showed the detection of the target bands in the mutant’s genome, confirming the successful acquisition of an *FpTox2* gene mutant (Figure 1C). The fragments, including the *FpTox2* gene promoter region and the entire open-reading frame, were amplified via PCR and transformed into yeast cells, together with the Xhol enzyme digestion vector pFL2. The constructed vector was transformed into the deletion mutation of the *FpTox2* gene using the protoplast transformation method. The transformed strain was verified using *FpTox2*-5F/6R primers, and a target fragment of the same size as the wild type’s was amplified, which proved that an *FpTox2* complementary strain (Δ*FpTox2*-C) had been successfully constructed (Figure 1D).

### 3.2. Tox2 Was Involved in Mycelial Growth and Conidiation

To investigate the regulatory role of FpTox2 in the mycelial growth of *F. pseudograminearum*, we cultured the wild-type 2035, Δ*FpTox2* mutant, and Δ*FpTox2*-C strains on PDA plates for 3 days. Mycelial plugs (5 mm diameter) were collected from the margins of the colonies and inoculated in triplicate onto a fresh PDA medium (Figure 2A). After 3 days of incubation, the colony diameters were measured and photographed. The Δ*FpTox2* mutant exhibited significantly reduced growth compared to both the wild-type and Δ*FpTox2*-C strains (Figure 2B), indicating that FpTox2 positively regulates vegetative growth in *F. pseudograminearum*.

We further examined FpTox2’s influence on conidiation and germination. Under identical induction conditions, the wild-type strain achieved conidiation yields of 2.53 × 10^6^/mL and 4.27 × 10^6^/mL in a CMC medium after 3 and 5 days of cultivation, respectively, while the Δ*FpTox2* mutant produced significantly lower yields (1.83 × 10^6^/mL and 0.92 × 10^6^/mL) (Figure 2C). Germination assays revealed 90% germination rates for the wild-type and Δ*FpTox2*-C strains versus 19% for Δ*FpTox2* after 6 h (Figure 2D). These findings demonstrate FpTox2’s positive regulatory role in both conidiation and conidial germination.

### 3.3. Analysis of Carbon Source Utilization and Abiotic Stress Responses

To examine FpTox2’s role in carbon source utilization by *F. pseudograminearum*, we compared the mycelial growth rates of the wild-type strain 2035, Δ*FpTox2* mutant, and Δ*FpTox2*-C strains on YSS media supplemented with various carbon sources (glucose, glycerol, sodium citrate, sodium succinate, sodium malate, and acetate; Figure 3A). Notably, Δ*FpTox2* displayed enhanced growth on glycerol compared to both the wild-type and Δ*FpTox2*-C strains. However, no significant growth differences were observed among the strains when utilizing succinate, citrate, glucose, malate, or acetate as carbon sources, indicating FpTox2’s specific regulatory role in glycerol metabolism.

For abiotic stress assessment, the strains were cultured on PDA for 3 days before being transferred to a CM medium. After 3 days of exposure to 10 mM concentrations of each stressor (Figure 3C), Δ*FpTox2* showed heightened sensitivity to NaCl and CaCl_2_ but reduced sensitivity to H_2_O_2_ relative to the control strains (Figure 3D). These findings demonstrate FpTox2’s significant involvement in *F. pseudograminearum*’s response to abiotic stresses.

### 3.4. Tox2 Is Necessary for DON Synthesis

To assess FpTox2’s impact on deoxynivalenol (DON) production, conidia from the wild-type, Δ*FpTox2* mutant, and Δ*FpTox2*-C strains were collected, standardized to equal concentrations, and cultured in 100 mL of TBI medium at 25 °C with 150 rpm shaking for 7 days. DON quantification was performed using a commercial ELISA kit (Wise Science and Technology Development Co., Ltd., Suzhou, China). The Δ*FpTox2* mutant exhibited a 93.18% reduction in DON production compared to both the wild-type and Δ*FpTox2*-C strains (Figure 4A), demonstrating FpTox2’s positive regulatory role in DON biosynthesis. To elucidate the molecular mechanism, we analyzed the transcript levels of *TRI5*, the key DON biosynthetic gene. Quantitative analysis revealed significantly reduced *TRI5* expression in the Δ*FpTox2* mutant, relative to the control strains (Figure 4B). These findings establish that FpTox2 modulates DON production through transcriptional regulation of the *TRI5* gene.

### 3.5. Tox2 Deletion Mutant Exhibits Reduced Virulence

To assess the virulence differences between the wild-type strain 2035 and Δ*FpTox2*, surface-sterilized wheat seeds (from the susceptible Shixin 828 variety) were germinated on moist filter paper in Petri dishes using 3% sodium hypochlorite. Following a 3-day PDA culture, mycelial plugs from the margins of *F. pseudograminearum* wild-type 2035, Δ*FpTox2*, and Δ*FpTox2*-C strain colonies were inoculated onto wheat coleoptiles. Pathogenicity assays revealed that both the wild-type and Δ*FpTox2*-C strains induced severe coleoptile infection (Figure 5A), while Δ*FpTox2* exhibited significantly reduced disease severity (Figure 5B), as quantified using disease index measurements.

## 4. Discussion

The killer toxin, a secretory protein, inhibits target cell growth and induces cell death by blocking voltage-gated calcium channels and calcium-dependent signaling pathways in target organisms [28,29,30,31]. Consequently, this secreted protein represents a potential target for fungicides in preventing and controlling plant fungal diseases [28]. In this study, we characterized the role of FpTox2, a previously unstudied KP4-like (KPL4) homologous protein, in *F. pseudograminearum*, a highly pathogenic soil-borne fungus responsible for Fusarium crown rot in cereal crops. Our findings demonstrate that FpTox2 plays crucial roles in multiple facets of fungal biology, including growth, asexual development, stress responses, and virulence.

Killer protein 4 (KP4) was initially identified in *Ustilago* as UmKp4, with homologous proteins now found across diverse fungal species and one moss species [33,35]. In *Ustilago maydis*, KP4 exhibits dual functionality: it eliminates sensitive smut fungi while suppressing Fusarium growth and plant root development through calcium absorption inhibition [35]. KP4 homologs in *F. graminearum* also mediate ecological interactions, with a particularly significant role in antagonizing *Trichoderma* biocontrol fungi. During coculture, Fgkp4l-3 and Fgkp4l-4 participate in antagonist distance recognition, with all Fgkp4l genes showing activation upon physical fungal contact, suggesting KP4L’s role in interspecific interactions within plant tissues [28]. Furthermore, during fungal infection of wheat seedlings, all four *FgKP4L* genes were expressed, and *FgKP4L* gene cluster deletion mutants exhibited significantly reduced virulence in regard to seedling blight on the susceptible wheat cultivar Ning7840 [32]. Notably, three KP4 homologs (FGSG_00060/Tox1, FGSG_00061/Tox2, and FGSG_00062/Tox3) exhibited differential expression patterns in infected wheat, with FGSG_00060 showing exclusive upregulation in wheat ears, demonstrating tissue-specific regulation [48]. However, our study identified FpTox2, a KP4 homolog in *F. pseudograminearum*, which showed no detectable induction during wheat coleoptile infection (data to be published separately).

Although the pathogenic role of KP4-related genes has been elucidated, their pathogenic mechanisms in plant pathogenic fungi remain unclear. In our study, deletion of FpTox2 in *F. pseudograminearum* led to (1) significantly impaired radial growth compared to that of the wild-type (Figure 2A,B), confirming this gene’s positive regulatory role in vegetative growth, and (2) the complete loss of the fungi’s conidiation capacity under induced conditions, along with markedly reduced conidial germination rates (Figure 2C,D), revealing this gene’s essential function in both processes. These results highlight FpTox2’s crucial role in fungal asexual reproduction, a key determinant of pathogen dissemination and survival in agricultural environments.

The *FpTox2* deletion mutant also exhibited altered carbon source utilization and stress response profiles. Specifically, the mutant showed enhanced growth on glycerol but no significant differences in its growth on the other carbon sources tested, suggesting a specific regulatory role for FpTox2 in glycerol metabolism (Figure 3A,B). Additionally, the mutant displayed increased sensitivity to NaCl and CaCl_2_ but reduced sensitivity to H_2_O_2_, indicating the complex role of FpTox2 in abiotic stress responses (Figure 3C,D). These observations highlight the multifaceted roles of FpTox2 in fungal physiology and adaptation to different environmental conditions.

Most importantly, our study demonstrates that FpTox2 is essential for achieving full virulence in *F. pseudograminearum* (Figure 5). The deletion mutant exhibited significantly reduced virulence in wheat coleoptile infection assays, with a markedly decreased disease severity compared to that of the wild-type strain. Furthermore, we found that FpTox2 positively regulates the production of deoxynivalenol (DON), a potent mycotoxin known to contribute to fungal virulence (Figure 4A). Transcriptional analysis revealed that FpTox2 modulates DON biosynthesis by upregulating the expression of *TRI5*, a key gene involved in the trichothecene pathway (Figure 4B). Although FpTox2, a KP4 homolog, lacks a DNA-binding domain and is not a transcription factor, it may regulate DON biosynthesis through direct interaction with specific transcription factors, thereby altering their affinity for the *TRI5* gene promoter. The Δ*FpTox2* mutation diminished the transcriptional activation capacity of these factors, leading to reduced *TRI5* gene transcription and a consequent downregulation of DON production. These findings establish a direct link between FpTox2 and fungal virulence, mediated through its effects on mycotoxin production.

## 5. Conclusions

This study provides novel insights into the functions of FpTox2 in *F. pseudograminearum*, revealing its critical roles in fungal growth, asexual development, stress response, and virulence. Its deletion severely compromises fungal fitness and pathogenicity, highlighting its potential as a target for managing Fusarium crown rot in cereals. These findings bridge the knowledge gaps between secondary metabolite genes and their mechanistic roles in hemibiotrophic fungal virulence and could inform the development of novel strategies for disease control in cereal crops. Future studies focusing on the interaction between FpTox2 and other virulence factors, as well as the exploration of potential targets for genetic manipulation to reduce fungal virulence, will further elucidate the complex mechanisms governing Fusarium crown rot and contribute to the development of sustainable agriculture.

## Figures and Tables

**Figure 1 cimb-47-00714-f001:**
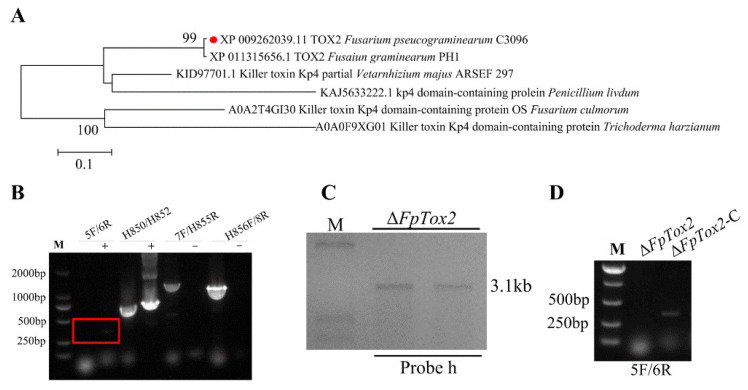
Phylogenetic analysis of FpTox2 and generation of its deletion mutant and complemented strains of *F. pseudograminearum*. (**A**) Phylogenetic analysis of FpTox2 from *F. pseudograminearum* (red circle) and homologous proteins from other fungi using MEGA 5. (**B**) PCR verification of *FpTox2* replacement mutants using four primer pairs, namely *FpTox2*-5F/6R, H850/H852, *FpTox2*-7F/H855R, and H856F/*FpTox2*-8R. The band amplified from the wild type strain is located within the red rectangle, whereas Δ*FpTox2* failed to amplify band of the same size (M, DNA marker; +, positive control; −, negative control). (**C**) Using hygromycin gene probe h to verify DNA of *FpTox2* deletion mutant through Southern blots. (**D**) PCR verification of complementation using *FpTox2*-5F/6R primers (Δ*FpTox2*-C, complemented strain).

**Figure 2 cimb-47-00714-f002:**
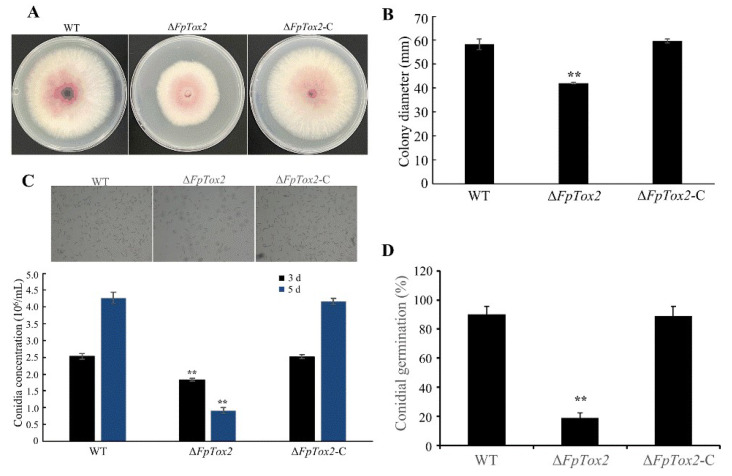
Effect of FpTox2 on the mycelial growth rate, conidiation, and conidial germination of the wild-type strain 2035 (WT), FpTox2 deletion mutant (Δ*FpTox2*) strain, and complemented strain (Δ*FpTox2*-C). (**A**) For 3 days, 5 mm mycelial plugs from the WT, Δ*FpTox2*, and Δ*FpTox2*-C were grown on PDA plates. (**B**) Determination of the colony diameter of the WT, Δ*FpTox2*, and Δ*FpTox2*-C after 3 days of incubation at 25 °C. (**C**) Conidiation of the WT, Δ*FpTox2*, and Δ*FpTox2*-C quantified using a hemacytometer after incubation in 50 mL of a CMC medium for 3 and 5 days, respectively. (**D**) The germination of the conidia was observed and recorded under a microscope after the strains were cultured for 6-8 h on an agar plate containing 0.2% glucose. There were 3 replicates per treatment, and the experiment was repeated 3 times. Asterisks on the bars represent statistically significant differences between the wild-type and tested strains (**, *p*<0.05; *t* test).

**Figure 3 cimb-47-00714-f003:**
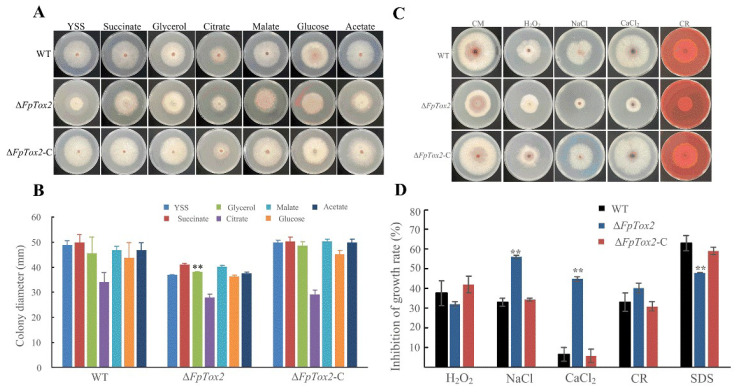
Effect of FpTox2 on carbon source utilization and the sensitivity to abiotic stress responses. (**A**) The mycelial growth of the wild-type strain 2035 (WT), *FpTox2* deletion mutant (Δ*FpTox2*), and complemented strains (Δ*FpTox2*-C) on a YSS medium containing 0.05 M of succinate, glycerol, citrate, malate, glucose, or acetate was determined after 3 days of incubation. (**B**) Determination of the colony diameters of the WT, Δ*FpTox2*, and Δ*FpTox2*-C on the YSS media containing different carbon sources after 3 days of incubation. (**C**) The WT, Δ*FpTox2*, and Δ*FpTox2*-C were grown on a CM medium supplemented with H_2_O_2_, NaCl, CaCl_2_, Congo Red (CR), or SDS for 3 days. (**D**) Statistical analysis of the strains’ inhibition rates after 3 days of incubation on the CM media treated with different stressors. There were 3 replicates per treatment, and the experiment was repeated 3 times. Asterisks on the bars represent statistically significant differences compared to the wild-type and complementary strains (**, *p*<0.05; *t* test).

**Figure 4 cimb-47-00714-f004:**
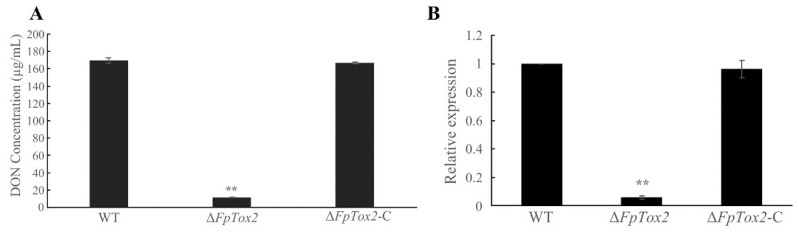
We quantitatively analyzed the DON production and expression of the DON synthesis-related gene *TRI5* in the wild-type strain 2035 (WT), *FpTox2* deletion mutant (Δ*FpTox2*), and complemented strain (Δ*FpTox2*-C). (**A**) The conidia of the WT, Δ*FpTox2*, and Δ*FpTox2*-C were incubated in a TBI medium for 7 days to determine the DON yield. (**B**) The *TRI5* expression level of each strain was determined in a TBI liquid medium after 3 days of incubation in darkness. There were 3 replicates per treatment, and the experiment was repeated 3 times. Asterisks on the bars represent statistically significant differences compared to the wild-type and complementary strains (**, *p* < 0.05; *t* test).

**Figure 5 cimb-47-00714-f005:**
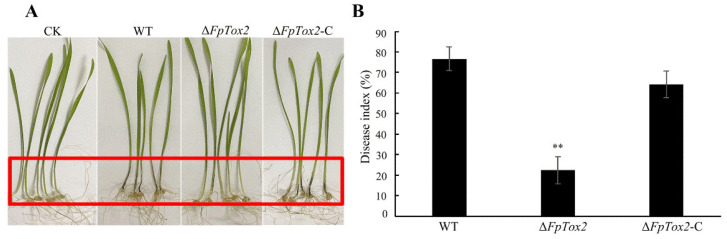
Comparison of the virulence of the wild-type, *FpTox2* deletion mutant, and complemented strains. (**A**) The stem bases were inoculated with mycelial plugs of the wild-type strain 2035 (WT), *FpTox2* deletion mutant (Δ*FpTox2*), and complemented strain (Δ*FpTox2*-C) and examined after 7 days of incubation. The red rectangle shows the affected area (CK denotes the control group of wheat plants subjected to mock inoculation procedures). (**B**) A statistical analysis was performed on the disease index of the WT, Δ*FpTox2*, and Δ*FpTox2*-C strains, which was calculated based on the different disease grades and observations of the infected wheat stem bases. Three biological repeats were performed. Asterisks on the bars represent statistically significant differences compared to the wild-type and complementary strains (**, *p* < 0.05; *t* test).

## Data Availability

The protein sequences analyzed in this study are available in public databases (National Center for Biotechnology Information-NCBI/Uniprot).

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
