# Peer review of "The TOX2 Gene Is Responsible for Conidiation and Full Virulence in Fusarium pseudograminearum"

_cimb, 2025, doi:10.3390/cimb47090714_

Round 1
Reviewer 1 Report
Comments and Suggestions for Authors
Comments
Abstract
(1) Line18-19: This pathogen infects and colonizes host tissues through the production of specific secondary metabolites (SMs). This sentence about the infection mechanism is not rigorous enough and it is suggested to be revised.
(2) Line 27 FpTox2 is crucial for plant growth?Please confirm whether there are spelling mistake of word plant.
Introduction
(1) Line46-47 The reference 9 cited in this sentence are inappropriate. It is recommended to replace them with more appropriate one.
(2) Line98 Please note that the gene Tri5 should be written in italics and the entire text should be thoroughly checked.
Methods
(1) Line 114-115. Has the whole genome of F. pseudograminearum strain 2035 been sequenced? It is recommended that the authors attach the NCBI number of FpTox2 gene in the text, or show the complete sequence of FpTox2 in the supplementary file.
(2) Line 153. “0.2 g/ml” should be changed to “0.2 g/mL”.
(3) Line164-165. Please indicate the temperature and humidity information used in the pathogenicity assay.
(4) Line173. (TBI)medium. This place is missing Spaces. Similar situations occur many times in the article, please revise accordingly.
(5) Line176. Please list the product number of commercial detection kit.
(6) Line180. stain?: Spelling mistake. 12h?: Space missing.
Results
(1) In line 191,192,201,202,204,215,220,223,229,231,247,257,274,277,282,295,298, 299,301,346.The word "FpTox2" should be written in italics. Please check the full text for similar writing omissions.
(2) In line 196. For PCR amplification, the length information of the fragment obtained In the wild-type strain needs to be indicated.
(3) Line 218. The capitalization of the first letters of the words in title 3.2. should be consistent with that of other titles.
(4) Line 228. Please revise the grammar of this sentence to make it more standard.
(5) Line 236.wild-type 2025? Perhaps there is a spelling mistake. It should be wild-type 2035.
(6) Line 237. 5mm?: Space missing.
(7) Line 250-252: This part of the text description does not match the picture. It is recommended to describe the results more accurately.
(8) In Figure 3 B, this bar chart should have a significant difference analysis and be marked with asterisks accordingly.
(9) In line 272. The capitalization of the first letters of the words in title 3.4. should be consistent with that of other titles.
(10) In line 294. The statement of “lost virulence” may not be rigorous, it is suggested to modify the wording.
(11) In line 307-308. Please check the grammatical errors in this sentence and correct them.
Discussion
(1) Line 339. F. pseudograminearum should be written in italics.
(2) Line 359-361. Transcriptional analysis revealed that FpTox2 modulates DON biosynthesis by upregulating the expression of TRI5. It is an interesting regulatory mechanism. It is suggested that the author present their own viewpoint and give an in-depth discussion on why FpTox2 affects the expression of the TRI5 gene, directly binding to the promoter? Or affect the activity of other transcription factors? or etc...
Comments on the Quality of English LanguageThe overall manuscript is well-written and the contents can be accurately presented. The grammar of some sentences can be further improved.
Reviewer 2 Report
Comments and Suggestions for Authors
The manuscript titled “The TOX2 Gene is responsible for conidiation and full virulence in Fusarium pseudograminearum” describes the functional role of FpTox2, a gene related to secondary metabolites in F. pseudograminearum.
The FpTox2 deletion mutant showed significantly reduced radial growth compared to the wild-type strain of F. pseudograminearum. Interestingly, the mutant completely lost conidiation capacity under induced conditions. Additionally, while showing decreased sensitivity to sodium dodecyl sulfate, a cell membrane inhibitor, the mutant demonstrated increased susceptibility to NaCl, a metal ion stressor. More importantly, pathogen virulence was significantly attenuated in wheat stem base infections after FpTox2 deletion.
It was demonstrated that FpTox2 regulates virulence by affecting deoxynivalenol production in F. pseudograminearum. In conclusion, FpTox2 is essential for plant growth, sexual and asexual development, and abiotic stress responses in F. pseudograminearum.
The manuscript is well-written and will be of high interest to international research community.
Reviewer 3 Report
Comments and Suggestions for Authors
The article by Sen Han , Shaobo Zhao , Yajiao Wang , Qiusheng Li , Mengwei Sun , Lingxiao Kong , Xianghong Chen , Jianhai Gao and Yuxing Wu «The TOX2 Gene is responsible for conidiation and full virulence in Fusarium pseudograminearum» is devoted to the study of one of the most important groups of fungal diseases – fusarium diseases, which leads to significant annual losses in the harvest of major food crops.
The article has a number of shortcomings that need to be addressed.
Introduction:
The authors explain in sufficient detail what prompted the authors' interest in this type of fungal pathogens, Fusarium pseudograminearum. However, it would be nice to confirm the relevance of the problem with numeric data. Is there an increase in damage to crops caused by Fusarium? Тhe volume of annual crop losses from this disease. Are there any positive developments in the fight against this type of plant disease? What are the crop losses due to Fusarium damage to economically significant crops? Which crops are most vulnerable to the effects of Fusarium pathogens.
Line 50
Let me disagree that secondary metabolites are not important for the survival of the organism. («are non-essential for fungal survival»), in this case, Fusarium pseudograminearum. Secondary metabolites make environmental adaptation possible . The level of their synthesis can vary significantly, which gives the effect of adaptation.
Materials and methods
2.1. Strains and growth conditions
The authors should specify in this section how many mutant lines were used and how they were obtained. How has the wild Fusarium line been confirmed to belong to Fusarium pseudograminearum?
2.3. Analysis of growth rate, conidiation and conidial germination rate
Lines 142-144
It is not entirely clear how the authors estimated the germination rate. The method of growing colonies is clear, but how the calculation was carried out - not.
It is necessary to specify the type and brand of the microscope used in the experiment.
Section 2.4. Utilization analysis of different carbon sources
Do I understand correctly: the authors estimated the completeness of assimilation of carbon sources by the size (diameter) of the colonies obtained on the substrate with the addition of various carbon sources?
2.5. Stress assay
The authors need to describe what types of stress they have chosen to study and why.
2.6. Plant infection assay
The authors need to describe the scale by which they assessed the infestation level of wheat seedlings with Fusarium.
2.7. Determination of DON production
Line 176
Give at least a brief description in the text of the principle of quantification of DON using a commercial detection kit
The "Results" section
In Figure 1, the designation M is entered, decipher it. In Figure 1B, mark where the wild line is represented, where the mutant lines are.
3.2. Tox2 was involved in Mycelial Growth and Conidiation
Line 220
Which lines were used in the experiment as a complemented strain? There is an explanation in the caption, but not in the text.
3.3. Analysis of carbon sources utilization and abiotic stress responses
Lines 254-260
The description of the types of stress should be duplicated in the section "Materials and methods".
Lines 250-253
The authors are confident that the visible difference was only in the utilization of glycerol.
In Figure 3B, a visible difference between WT, ΔFpTox2-C and Δfptox2 is observed in the utilization of succinate, malate, acetate, and glucose.
In the "Results" section, the authors describe the data obtained for three lines: the wild WT line, two mutant lines - ΔFpTox2-C and Δfptox2. The authors use various synonyms when describing: control lines, complemented strains. It is better to use uniform terminology to avoid confusion.
3.5. Tox2 deletion mutants lost virulence
Describe the principle of calculating the disease index in the section "Materials and methods".
In Figure 5A, select the affected area using a rectangle or something else, because the drawings are quite small and the area affected by the fungal pathogen is poorly distinguishable.
Describe the fusarium infection of wheat seedlings in the text using the scoring system, which is mentioned in the relevant section of "Materials and Methods" to confirm your conclusions.
It is better to highlight the conclusion separately.
Round 2
Reviewer 3 Report
Comments and Suggestions for Authors
The authors made corrections to the text and answered questions.
The article can be accepted for publication.